

# 1   GPR and IRT Tests in two Historical Buildings in Gravina in Puglia

Loredana Matera[1], Raffaele Persico[1], Edoardo Geraldi[1], Maria Sileo[1], Salvatore Piro[2]
1: Institute for Cultural and Monumental Heritage IBAM-CNR
2: Institute of Science for Knowledge, Conservation and Use of Cultural Heritage ITABC-CNR
**Abstract**
*This paper describes a noninvasive investigation conducted in two important churches, namely the*
*Cathedral of Santa Maria Assunta and the church Santa Croce, both placed in Gravina in Puglia*
*(close to Bari, southern Italy). The church of Santa Croce, now deconsecrated, lies below the*
*Cathedral. Therefore, indeed the two churches constitute a unique building body. Moreover, below*
*the church of Santa Croce there are several crypts, only partially known. The prospecting was*
*performed both with a pulsed commercial GPR system and with a prototypal reconfigurable*
*stepped frequency system. The aim was twofold, namely to achieve some information about the*
*monument and to test the prototypal system. The GPR measurements have been also integrated with*
*an IRT investigation performed on part of the vaulted ceiling of the church of Santa Croce, in order*
*to confirm or deny a possible interpretation of some GPR results.*
**Keywords**: Ground Penetrating Radar; Cultural Heritage; IRT.

## 20   1. Introduction

Noninvasive diagnostic technologies, and in particular Ground Penetrating Radar (GPR) and Infra-
Red Termography (IRT), are important tools to study architectural and monumental heritage. The
use of non-destructive testing (NDT) arises from the exigency not to damage the probed structure,
which is a particularly important requirement for cultural heritage. Noninvasive sensing makes
possible to achieve both historical and structural information about the building at hand (Masini et
al., 2010; Utsi E., 2010; Calia et al., 2012, Grinzato et al, 2002, Geraldi et al., 2003, Carlomagno et
al., 2011, Geraldi et al., 2016). In particular, it has been exploited to document the state of damage
of masonries (Masini et al., 2010), columns and pillars (Binda et al., 2003; Leucci et al., 2011) or
even statues (Kadioglu and Kadioglu, 2010, Sambuelli et al., 2011). NDT also represents an
important tool to detect the presence of ancient tombs (Cardarelli et al, 2008), crypts, cavities
(Piscitelli et al., 2007, Persico et al., 2014), archaeological structures (Goodman and Piro, 2013) and
also a tool useful for the study of murals and frescoes (Pieraccini et al 2006).
The GPR technique is based on the scattering of the electromagnetic wave radiated by a
transmitting antenna and impinging on any buried anomaly, that are scattered along all the
directions and in particular along the direction of the receiving antenna, that gathers a small share of





this scattered energy. Depending on the characteristics of the medium and on the frequency of the
antennas, the GPR technique can investigate usually the first meters of depth (let say about from 1
to 7 meters) with a resolution also depending on the characteristics of the medium and on the
frequency of the antenna, of the order of one half of the internal wavelength, which means of the
order of 1-40 cm. Moreover, there is some degradation of the resolution vs. the depth.
The IRT technique (Maldague, 2001) is based on the thermal radiation and on the heat transfer
mechanism that occur between the target's surface and the thermal camera sensor (mainly emission
of the target and reflection by the surroundings). The properties that lead the thermo-physical
phenomena (conduction and radiation) between the target, its surface and the surroundings
(boundary conditions) are thermal properties as conductivity, diffusivity, effusivity and specific
heat capacity, spectral properties as emissivity, absorption, reflection, transmission and further
physical properties as volumetric mass density, porosity and parameters defining the hygrometric
conditions. An infra-red camera measures the thermal radiation coming from the material under
investigation and renders the image of the surface area in relation to a temperature scale. The
temperature mapping of the surface area is strictly connected to the ongoing thermal transient state
generated actively from an exterior heat source or passively inducted from the variable
environmental physical conditions (solar irradiation, wind, diurnal fluctuation of temperature,
humidity, etc.) as well as from the medium physical characteristics (materials, texture, degradation
processes, etc.).
In this paper, we present some GPR and IRT results achieved in two churches in the town of
Gravina in Puglia, in the outskirts of Bari (Apulia Region, southern Italy), namely the Cathedral of
Santa Maria Assunta and the underlying church of Santa Croce.
In fig. 1 the geographical location plus some images of the two churches are provided.

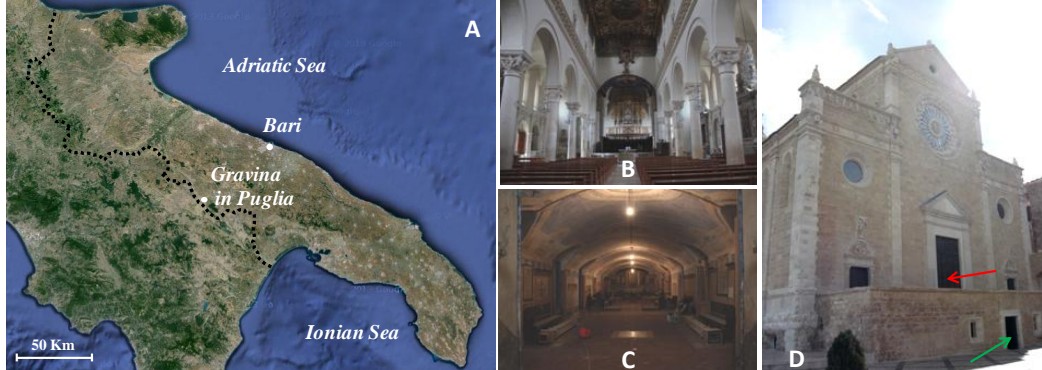


Fig. 1 : (A): Geographical location of Gravina in Puglia; (B): the central nave of the Cathedral; (C): the central nave of
the church of Santa Croce; (D): the external façade of the building with the entrance of the Cathedral (upper red arrow)
and of the church of Santa Croce (lower green arrow).



This building constitutes a natural test site for experimenting the capabilities of a GPR system to
penetrate into a multilayered structure. In particular, we will show how some of the main features of
the church of Santa Croce are reproduced from GPR data taken on the floor of the Cathedral and we
will show images ascribable to a few crypts (some of which accessible and some of which not
accessible) below the church of Santa Croce, achieved from GPR data gathered on the floor of
Santa Croce.
Indeed, some historical sources (Lorusso et al., 2013) and some local rumors state the presence of at
least two levels below the church Santa Croce, and even a third level is hypothesized.
However, the reflection of the electromagnetic waves from a large cavity is in general quite strong,
and customarily it is quite hard to identify a second (large) cavity piled below a shallower (also
large) one, because the shallower one can mask the deeper one. With regard to the case history at
hand, in particular, we were unable to identify the crypts under the church of Santa Croce from the
floor of the upper-lying Cathedral, and so we easily infer that we were also unable to identify
further levels possibly placed below the crypts identified from the floor of Santa Croce.
The prospecting was performed with two different GPR systems, namely a pulsed Ris-Hi mode
manufactured by IDS-Corporation, equipped with a double antenna at 200 and 600 MHz and a
prototypal stepped frequency reconfigurable system (Persico and Prisco, 2008). The prototypal
system was implemented within the research project Aitech (http://www.aitechnet.com/ibam.html)
by the Institute for Archaeological and Monumental Heritage IBAM-CNR together with the
University of Florence and the IDS corporation, and underwent several tests in different situations
within a Ph.D. course handled in collaboration between the University of Bari and the Institute for
Archaeological and Monumental Heritage IBAM-CNR (Persico et al., 2014).
Finally, after examining the GPR results, we deemed worth performing also another non invasive
test performed with an infra-red camera on part of an internal wall of the cylindrically vaulted
ceiling of the main nave of the church of Santa Croce, in order to have further elements for a more
reliable interpretation of the GPR data.
The paper is organized as follows. In the next section, a brief historical description of the two
monuments is provided. In section 3, a brief description of the reconfigurable system is given. In
section 4, the GPR measurement campaigns are described and the main results are shown. In
section 5 the infra-red investigation is addressed. Conclusions follow in section 6.

**2. Historical outlook**




Historical sources (D'Elia P., 1975) report that the presence of two overlapped or multilayered
churches, as the case of the Cathedral of Santa Maria Assunta and the church of Santa Croce in
Gravina in Puglia, is common in several Apulian churches (e.g. in the towns of Bari, Trani and
Bisceglie). The most probable reason of such architecture is that there was a previous church
subsequently incorporated in the upper-lying one, because the demolishing of the previous church
was discouraged due to the sacredness of the place.
With regard to the building at hand, the construction of the Cathedral of Santa Maria Assunta, in
Apulian Romanesque style, dates back to 1092 (Gelao C., 2005).
During the centuries, it has been reconstructed with many difficulties and interruptions due to the
scarcity of funds and due to several earthquakes that afflicted the town of Gravina. Only few rests
of the ancient Cathedral (the rose window, some external frames and some capitals of the arches
that from the aisles that lead into the presbytery) have been preserved.
Some authors (Gelao C., 2005) report the that Cathedral of Santa Maria Assunta was built over the
church of Santa Croce also in order to overcome difficulties related to the topographic morphology
of the site, which would have compelled the initial constructors to build the church of Santa Croce
within a natural depression, locally called *gravina*.
Nowadays, the Cathedral has two entrances, one of which is located on the southern facade, in
Benedetto XIII Square whereas the other one (the portal shown in fig. 1D) is placed on the main
façade with a few stairs above the ground level, in front of a rectangular open space.
A big chapel protrudes from the left hand aisle and stands sheer on a deep *gravina*. This big chapel
was built in the first half of the seventeenth century with two floors. At the lower floor, in the
church of Santa Croce, the oratory is placed. At the upper floor, in the Cathedral, the Blessed
Sacrament chapel is placed. A further prestigious architectural element is the *bell tower* (Lorusso et
al., 2013).
Inside the Cathedral there are three naves, separated from each other by fourteen columns linked
together by round arches. The ceiling of the nave is wooden, carved and gilded according to the
Baroque style. The entire Cathedral covers an area of 50x20 square meters. The height of the nave
is 20.90 meters, while that of the aisles is 12.70 meters. More detailed information about the
Cathedral can be found in Lorusso et al., 2013.
The underlying church of Santa Croce is also constituted of a central nave and two lateral aisles,
divided by massive squared pillars with barrel vaults, and has three altars. The church was closed to
worship in 1958, and until then it had been used to bury corps underneath it. Therefore, some graves
were dug under the floor level (Lorusso et al., 2013). Nowadays, some frescos remain and twelve
graves (Lorusso et al., 2013), dating back to a period from the sixteenth to the nineteenth century.






### 3.  The prototypal stepped frequency system
The prototypal reconfigurable system exploited in the case history at hand is a stepped frequency
GPR whose frequency range extends from 50 MHz to 1 GHz. This range can be swept with a
frequency step optionally equal to 5 or 2.5 MHz. One of the main innovation points related to this
system is the fact that it is equipped with three couples of equivalent antennas. These three couples
are achieved from a unique physical couple of antennas, by means of two series electronic switches
(implemented by means of PIN diodes) displaced along the arms of the two antennas according to
the scheme of fig. 2. The three couples of equivalent antennas are achieved switching on and off the
PIN diodes, which provides a couple of "long antennas" if both the switches per arm are set on, a
couple of "medium antennas" if the external switches are set off and the internal switches are set on
and "short antennas" if both switches are set off. In other words, the switches implement two
subsequent equivalent cut of the arms of the antennas. We have seen that (with some site dependent
variations), the central frequencies are of the order of 120, 250 and 550 MHz for the long, medium
and short antennas, respectively. Moreover, as usual in GPR antennas, the bands of the equivalent
antennas are of the same order of the central frequencies.
Indeed, the problem of the cut of the arms is more complicated than it might seem, because the
detached parts of the arm provides necessarily a contribution both to the antenna pattern and to the
input impedance, because some induced currents flow on them. However, from a practical point of
view, there are two factors that mitigate the problem: the first one is the fact that the antennas are
bow-tie, and this makes the axis of the arms a direction of null of the pattern. Therefore, the
parasitic current developed on the (collinear) detached parts are expected to be tolerable. Quite
stronger induced currents, instead, might develop if the detached parts were parallel to the active
parts of the antennas, as it happens e.g. in the case of a Uda-Yagi antenna (Grajek et al., 2004). In
particular, the only component of the field that might induce meaningful currents on the detached
parts in our case is the radial one, that vanishes proportionally to the inverse of the square of the
distance from the gap (unlike the longitudinal component, that attenuates as the inverse of the
distance). A second factor that mitigates the effect of the cuts is the fact that, when the short
antennas radiate and receive the signal, the cut part of the arms are in their turn cut in two parts,
which prevents for the development of too strong parasitic currents.
In the case history at hand, we have gathered the entire available range of frequencies 50-1000 MHz
with each of the three equivalent antennas, then the data relative to each antenna have been filtered
into their own bands.



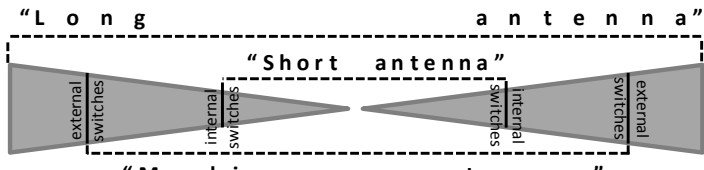



Fig. 2: Non-quantitative scheme for the reconfigurable antennas: the antennas are "short" if all the switches are set off,
they are "medium" if the internal switches are set on and the external ones are set off, they are "long" if all the switches
are set on.

Incidentally, the prototypal system allows to reconfigure not only the length of the arms of the
antennas but also the radiated power and above all the integration time (Noon, 1996) for each
radiated harmonic signal. However, these two further options are not of interest with regard to case
history at hand and we will skip over them here. For the interested reader, some more details about
the reconfiguration of the integration times are provided in (Persico et al., 2015).

**4. GPR results**
**4a: Results in the Cathedral**
The layout of the B-scans gathered in the Cathedral of Santa Maria Assunta is shown in fig. 3. A
total number of 14 B-scans ($L_1$-$L_{14}$) were acquired, eight of which parallel to the main nave and six
orthogonal to it. The length of the six B-scans recorded in the two aisles ranged from 30 m to about
32 m (namely the B-scans $L_1$-$L_3$ in right hand aisle and the B-scans $L_6$-$L_8$ in the left hand aisle), the
length of the B-scans in the main nave ($L_4$ and $L_5$) was about 24 m. The length of the B-scans $L_9$-
$L_{10}$, close to the entrance of the Cathedral, was about 14 m and the B-scans close to the transept
($L_{11}$-$L_{14}$) were about 18 m long. The church is in use and we did not have the permissions for
gathering a complete grid of data. Moreover, we have represented the projection of the B-scans $L_4$,
$L_7$ and $L_{13}$ on the map of the underlying church of Santa Croce shown in fig. 4. This will make
easier to understand the following part of this section. We have achieved this representation from
the quantitative maps of the two churches, that can be quite precisely "hanged" to each other
through the above described chapel protruding on the *gravina*.




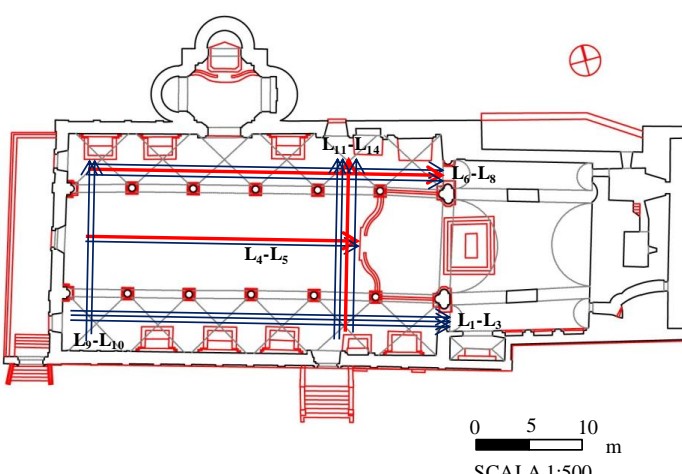


191           Fig. 3: Map of the Cathedral of Santa Maria Assunta with the layout of the gathered B-scans.

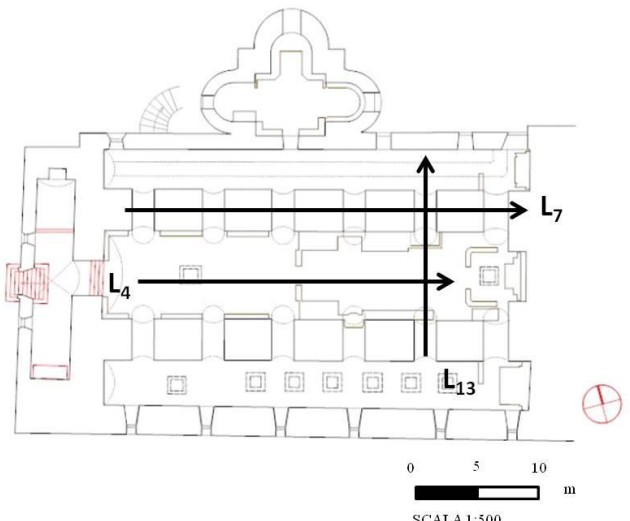


193         Fig. 4: Map of the Curch of Santa Croce, with the projections of the layouts of the B-scans $L_4$, $L_7$ and $L_{13}$ gathered on

194                        the floor of the Cathedral of Santa Maria Assunta.


Three of the B-scans, gathered with both the commercial and the prototypal systems, are shown in
figs. 5 and fig. 6. The data were processed with the ReflexW code (Sandmeier, 2011) through the
following steps: zero timing, two-dimensional filtering constituted by a subtracting average on 40
traces, variable gain vs. the depth and a further one-dimensional bandpass filtering. For the
collected data, no migration was carried out, because the anomalies looked for were indeed quite
large, and we have seen that this focusing procedure did not provide satisfying results. Probably,





this is due to the meaningful nonlinear effects related to the size and the complexity of the
scattering anomalies at hand (Persico et al., 2002).
The radar sections reported in the upper panels of figs. 5 and 6 (which refer to the B-scan $L_7$,
evidenced by a red arrow in figs. 3 and 4) have been achieved in the left hand aisle (looking toward
the altar) and refer to the data gathered with the pulsed and the prototypal system, respectively. The
recorded signals at 14 ns (see the yellow arrows in figs. 5 and 6) indicate the presence of six single
arches at the same time-depth and equally spaced at about 5.6 m from each other. Another signal at
40 ns (see the yellow arrows in fig. 5 and in fig. 6) is also present, even better visible than the
reflections at 14 ns. The first signals at 14 ns is related to the top of the vaults of the over-crossed
part of the ceiling of the church of Santa Croce, whereas the signals at 40 ns could be referred to the
relative underlying parts of the floor of Santa Croce. In particular, the six arches correspond to the
underlying arches of the lateral openings between the central nave and the lateral aisles of Santa
Croce, as can be understood from fig. 4.
From the diffraction hyperbolas, we have evaluated a propagation velocity of the electromagnetic
waves about equal to 0.12 m/ns. Therefore, the upper surface of the six vaulted feature is estimated
to be at about 0.85 m from the floor of the Cathedral, while the thickness of the structure is
estimated to be about 4 m, in good agreement with the ground truth visible from the church of Santa
Croce. Moreover, in the B-scan $L_7$ (but even more in the subsequent image of the B-scan $L_4$) very
superficial signals equally spaced are also present (red arrows in fig. 5 and in fig. 6). They are likely
to be ascribable to a welded steel mesh (with a step of about 50 cm) under the floor of the Cathedral
(that of course is not original). Below the time depth of 40 ns, several multiple signals are also
recorded.
The B-scans shown in the middle panels of figs. 5 and 6 have been acquired in the transept of the
Cathedral ($L_{13}$, marked with a red arrow in fig. 3 and evidenced also in fig. 4). In both radar
sections it is possible to notice the presence of a first flat ceiling placed at the time-depth of 16 ns,
about 3 m long, followed by a vaulted ceiling whose top is at 7 ns, about 6 m long, still followed by
a flat ceiling again at 16 ns about 4m long, followed by a further vaulted ceiling whose top is at 7
ns, about 2 m long. According to the evaluated propagation velocity, the depth of the top of the
vaults appears to be of the order of 0.40 m (but probably it is slightly larger of this value, for
engineering reasons) under the floor of the Cathedral, while the two flat ceilings appear to be at a
depth of about 1 m. Under this "comprehensive" curved-line-shaped reflector, there is a flattish
reflector at the depth of about 35 ns, that we interpret as the reflection from the floor of the church
of Santa Croce. Making use of the estimated propagation velocity of 0.12 m/ns before the curved-
line-shaped reflector, and making use of the propagation velocity in free space beyond it, we



estimate a thickness of 2.85 m under the two flat ceilings and a maximum height of the underlying
room of about 4.20 m under the top of the vaulted reflector. This is coherent with the height of the
underlying church of Santa Croce in the main nave and the lateral aisles as well as with the height
of the corridors that connect them to each other. In this B-scan also further buried structures are
recorded. In particular, likewise the previous B-scan, some superficial signals (red arrows in the
middle panel of figs. 5 and 6) and multiple signals below 35 ns are observed.
The third image, reported in the lower panels of figs. 5 and 6 is the B-scans $L_4$, achieved in the nave
of the Cathedral and marked with a red arrow in fig. 3 and put into evidence also in fig. 4. Two flat
reflections throughout the whole length of the radar sections at a time-depth of 6 ns and of 35 ns are
visible (they are indicated with yellow arrows in the lower panel of figs. 5 and 6). We estimate that
these signals are ascribable to the ceiling and to the floor of the underlying church nave. In this
image, the ceiling of the main nave of the church of Santa Croce appears to be flat because of
course the GPR path is parallel to the main nave. The time depth of 6 ns (quite close to the top of
the vaults at 7 ns visible in the previous panel) indicates that we have passed quite close to the top
of the underlying cylindrically vaulted ceiling. In the lower panels of fig. 5 and 6, also five
hyperbolic reflectors are visible, with their top at about 7 ns.
They are indicated with further yellow arrows in the figures. These reflectors are spaced about 5.75
m from each other, starting from the abscissas 0.5 m. We interpret these reflector as reinforcement
structures, developing in the direction orthogonal to the nave, or maybe keystones made of a
different material with respect to the surrounding bricks.
Also in this radar section the contribution of some shallow metal objects is visible, as well as some
signal probably arising from multiple reflections beyond the time depth f 40 ns.
To sum up, the B-scans shown in figs. 5 and 6 show that the two GPR systems provide results in
good agreement with each other. The main difference regards the contribution of the superficial
welded steel mesh, that appears less marked in the prototype radar sections.





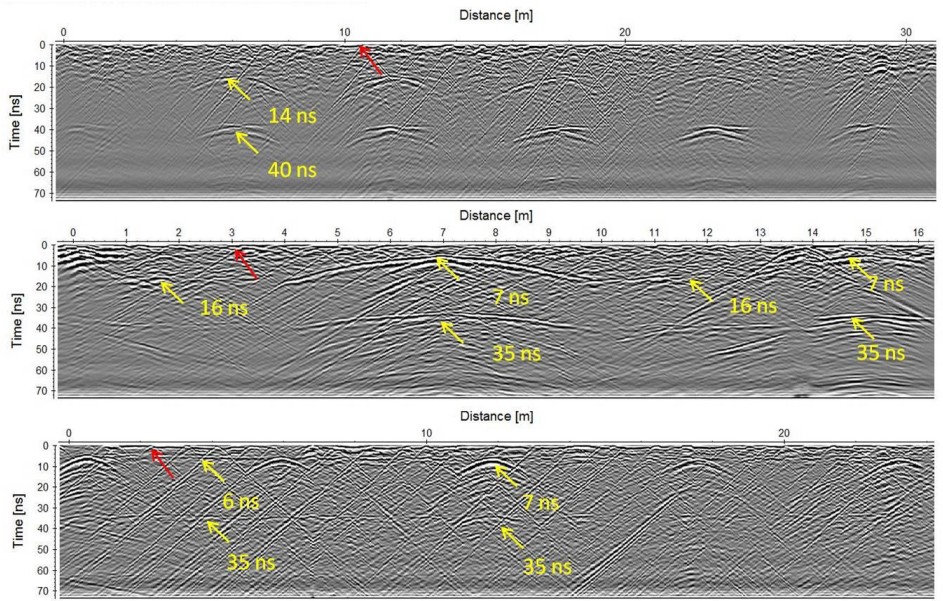

Fig. 5: Radar sections recorded with the antennas at 600 MHz of the commercial system in the Cathedral: $L_7$ (upper panel); $L_{13}$ (middle panel); $L_4$ (lower panel).

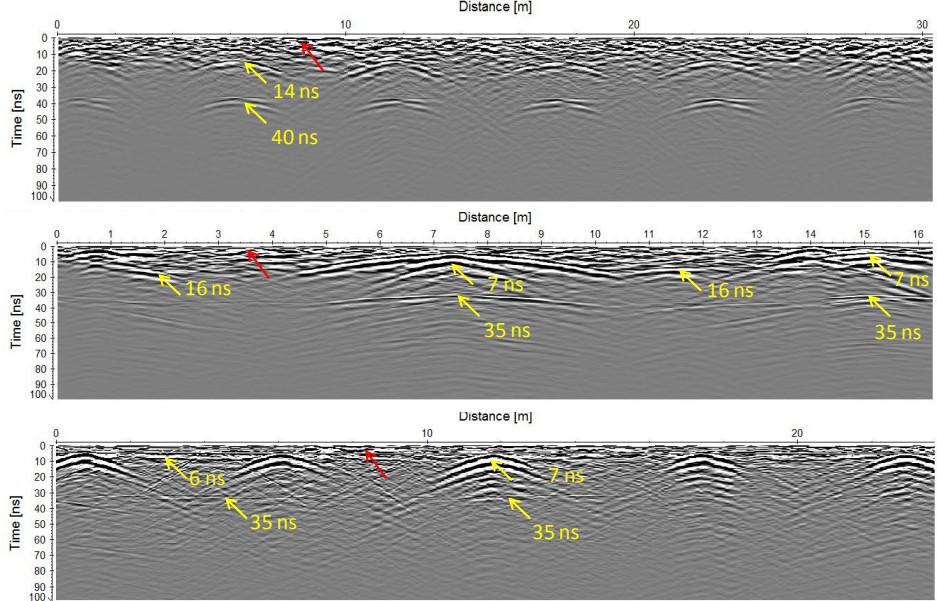

Fig. 6: Radar sections recorded with the "medium antennas" of the GPR-R prototype system in the Cathedral: $L_7$ (upper panel); $L_{13}$ (middle panel); $L_4$ (lower panel).

**4b: Results in the church of Santa Croce**





In the church of Santa Croce, three different areas, marked in fig. 7 with black rectangles and
labeled "Area1", "Area2" and "Area3", have been investigated. In the map of fig. 7, a known
hypogeal chamber tomb under the floor is represented too, by means of red contours. The chamber
tomb is composed of a main rectangular room and a lateral niche used for sepultures, connected to
the floor of Santa Croce with a straight staircase that leads to a marble manhole.

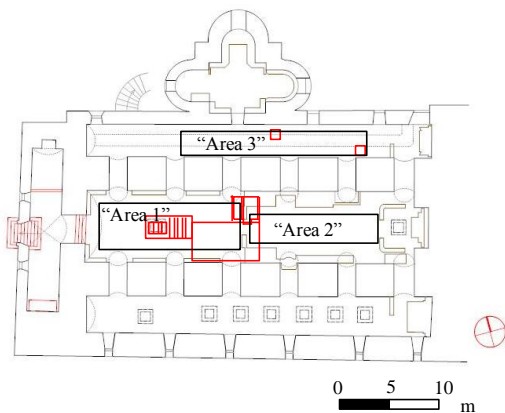

Fig. 7: Map of the church of Santa Croce with the three investigated areas and with the map of an underlying crypt and
of two manholes in red contour.
Other manholes are visible too in the lateral aisles, but only that relative to the represented chamber
is accessible and could be opened. For each area, an orthogonal grid of B-scans with interline step
of 50 cm was gathered with both the GPR systems at hand. Likewise the data gathered in the
Cathedral, also the data gathered in Santa Croce were processed using the ReflexW code.
The data of Santa Croce were migrated in time domain, making use of a propagation velocity of the
electromagnetic waves equal to 0.09 m/ns, retrieved on the basis of the diffraction hyperbolas.
From the processed B-scans, horizontal depth-time slices from 0 to 70 ns were retrieved, with time
windows of $\Delta t = 5.5$ ns. The most significant depth-time slices for both the GPR systems have been
reported in fig. 8. They refer to data collected with the antennas at 600 MHz for the pulsed system
and with the "medium antennas" for the GPR reconfigurable prototype system.
As it can be seen, the two GPR systems are in good agreement to each other, but the GPR
reconfigurable prototype depth-time slices provide better localized anomalies than those obtained
with the pulsed GPR device.





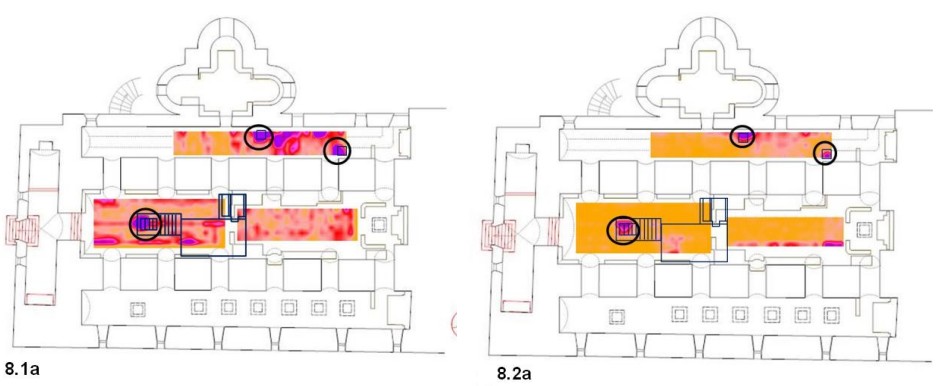

Slices at 5 ns: pulsed system (8.1a); stepped frequency system (8.2a)

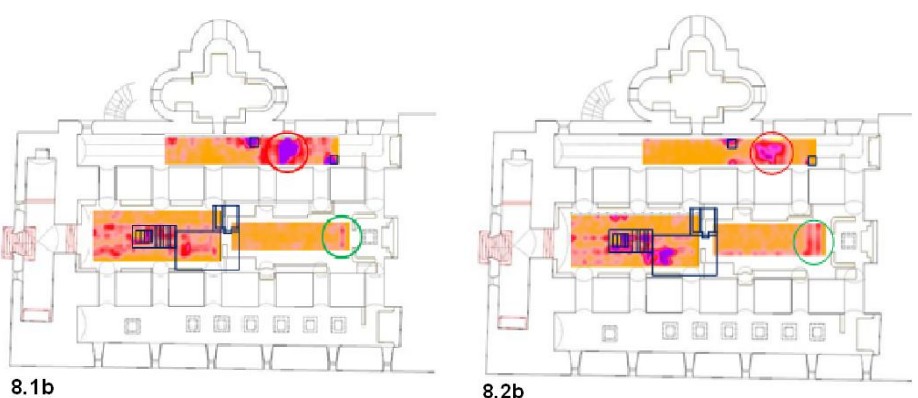

Slices at 15 ns: pulsed system (8.1b); stepped frequency system (8.2b)

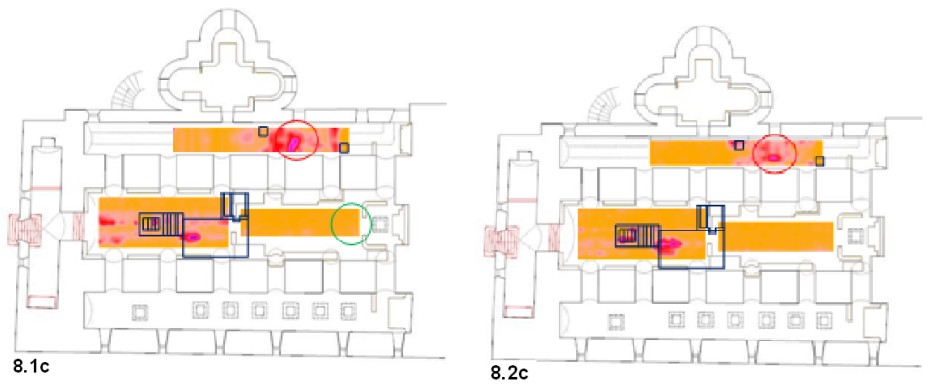

Slices at 35 ns: pulsed system (8.1c); stepped frequency system (8.2c)





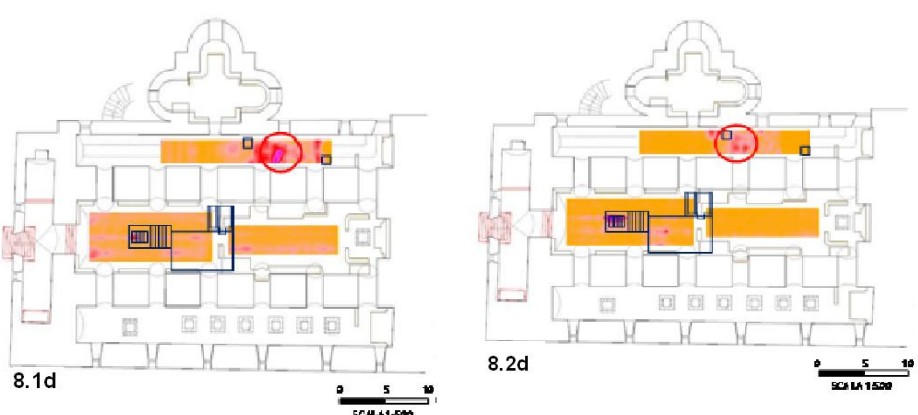

Slices at 65 ns: pulsed system (8.1d); stepped frequency system (8.2d)


Fig. 8: A comparison of the results achieved with: the antennas at 600 MHz for the pulsed system (fig. 8.1a-8.1d); the
"medium antennas" for the GPR-R prototype system (fig. 8.2a-8.2d).

With regard to Area 1, the prospected area covers a surface of 61 square meters, which correspond
to 38 B-scans in all, 10 of which parallel to the nave and 28 orthogonal to it. In Area 2, 30 B-scans
were recorded, covering an area of 29 square meters. With regard to Area 3, in the right hand aisle,
the surveyed area was about sized 33 square meters (5 B-scans parallel to the aisle and 34
orthogonal to it). It was not possible to prospect the entire aisle because of irremovable obstacles.
From the first depth-time slice at 5ns, corresponding to about 22 cm (fig. 8.1a-8.2a) we see that
both systems enounce the presence of shallow localized anomalies, which are three manholes. At
the depth of 15 ns (about 70 cm) a strong and large anomaly is identified in Area 3, probably
corresponding to a chamber tomb whose ancient entrance was trough the closest manhole on the left
hand side of the spot in Area 3. Still at 15 ns a strong anomaly in Area 1 is identified. It is probably
ascribable to the ceiling of the large known underlying chamber tomb. However, the spot has an
extension meaningfully reduced with respect to the actual extension of the prospected area over the
chamber. This is partially explained on the basis of the fact that the ceiling of the underlying
chamber is not flat. In particular, there is a cylindrically vaulted ceiling in the first part of the
chamber (i.e. the left hand part, closest to the steps that drive to the manhole in the map of fig. 7)
and a second part of the room where the ceiling is more than one meter deeper. At the depth of
about 35 ns, again two meaningful anomalies from Area 1 and Area 3 are visible. They might
correspond to the floors of the two chambers.
With regard to Area 2, the main identified anomaly is indicated with a green circle in fig. 8.1b and
fig. 8.2b. We interpret it as another possible tomb, smaller than the first one, whose entrance may be
the manhole on the final altar visible in the plans on figs. 7 and 8.

**5.  IRT results**
Beyond the features visible thanks to B-scans and the depth slices shown in the previous section,
there was some concern common to several longitudinal B-scans gathered along the short side of
the main nave of the church of Santa Croce. In particular, with reference to fig. 9, we can see a
central target at the time depth of about 30 ns that could be misinterpreted as the cross point of two
large diffraction hyperbolas, or maybe (which was still more concerning) the top of some buried
wall. In fact, this anomaly is visible along all the B-scans performed along the direction orthogonal
to the main nave of the church (Areas 1 and 2), and it is clearly visible with both systems.

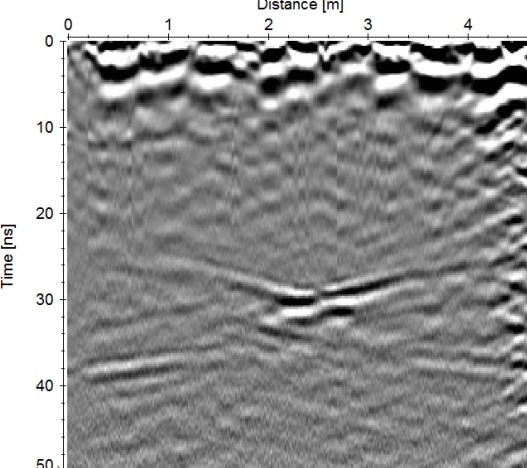


Fig. 9: B-scan along the short side of the main nave of the church of Santa Croce.

Indeed, there is a central sequence of old neon lights hanged to the ceiling. However, even if the
neon lights were alleged to be possibly the cause of the feature in fig. 9, the shape of the anomaly
was quite different from a diffraction hyperbola. Therefore, we have investigated the presence of
further possible long targets inserted in the wall (as e.g. internal pipes for the water or internal
electrical wires). To do this, the lateral walls and the vaulted ceiling were noninvasively probed
with a FLIR SC 660 infrared camera (FPA detector uncooled microbolometer operating in the
spectral range between 7.5 and 13 μm). In order to perform this measurement properly, the areas to
be investigated were preventively subjected to thermal stress with two halogen lamps.





Due to the presence of frescoes on the vaults, it was needed to control the entity of the thermal
stress, and this limited the thickness of the transient state inducted to the first millimeters of the
medium. However, this was not a strong drawback in the case at hand, because the pipe, if any, was
expected to be superficial. From the thermal images shown in fig. 10, we can appreciate anomalies
mainly ascribable to the discontinuities of the emissivity of the ceiling, in their turn ascribable to the
different pigments that characterize the frescoes (see panel C in particular). Further weaker
anomalies seem to be ascribable to the shallower level of the ashlars below the plaster (see panel
D). In particular, the IRT images do not show any evidence of intra-wall pipes or wires, so that they
make us exclude such a hypothesis.

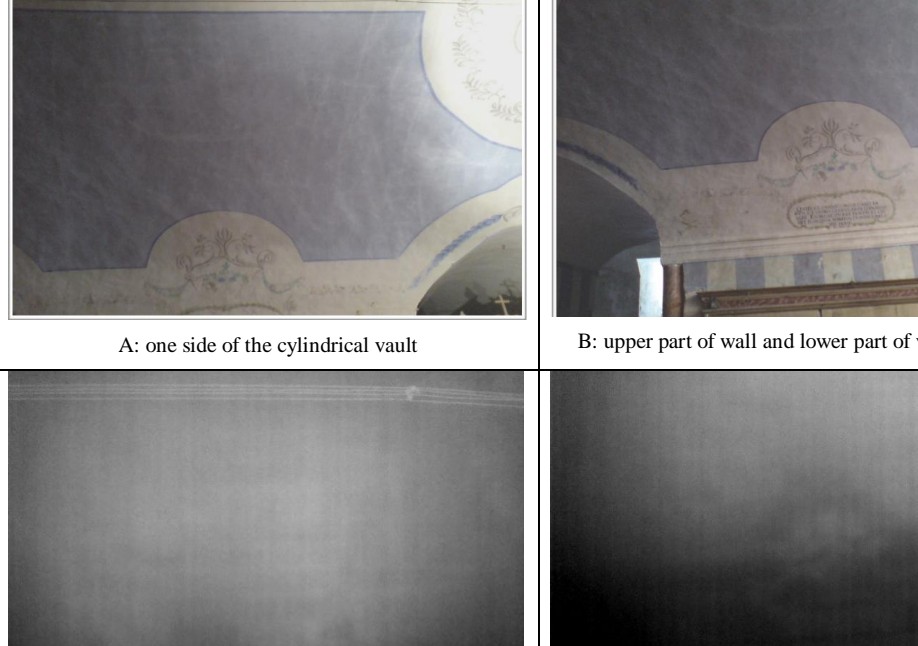

Fig. 10: The IR investigated areas in the nave of the church of Santa Croce (panel A and panel B); IR images relative to
the photo of panel A and B (panel C and panel D).

**6. Conclusions**
In this paper, the results of a non-destructive survey performed inside the Cathedral of Maria
Assunta Cathedral and the church of Santa Croce in Gravina in Puglia (Apulia, Italy) have been




shown. We have tested, in particular, an innovative stepped frequency GPR system vs. a traditional one, and have shown that it provided results fully comparable with those achieved from a traditional GPR system. An advantage of the prototype is that we have more bands at disposal in order to look for the best imaging of the targets of interest. This survey allowed us to verify that the main features of the church of Santa Croce were visible from the upper floor of the Cathedral of Maria Assunta, and allowed us to identify the presence of underlying structures below the church of Santa Croce. However, neither the pulsed system nor the prototypal stepped frequency were able to see chambers below the first buried level under the observation line. Therefore, the problem whether further levels are present below the crypts under the church of Santa Maria remains open.

**Acknowledgements**

The project Aitech, that has allowed the implementation of the reconfigurable GPR system, has been financed by the Apulia Region.

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
