# Peer review of "GPR and IRT Tests in two Historical Buildings in Gravina in Puglia"

_Geoscientific Instrumentation, Methods and Data Systems, 2016_

## Referee Comment (RC1) · Anonymous Referee #1 · 26 Jul 2016

The paper is almost ready for the publication but it is necessary still to work on the section 5 regarding IRT results. It is not clear what is the correlation and the added value provided by IRT with respect to the GPR measurements. At present, section 5 does not appear well linked to the other sections of the manuscript.

---

## Author Comment (AC1) · 23 Aug 2016

We have modified the IRT section accordingly, when we receive the other comments we will resubmit an updated version of the paper.
* * *

---

## Referee Comment (RC2) · Anonymous Referee #2 · 9 Sep 2016

The revised version has been improved, but the authors have not proved that the echoes (part of hyperbolas) visible on the Fig. 9 are not aerial echoes. I suspect, if I could superpose a hyperbola with a commercial software, that I would obtain a velocity of about 30 cm/ns : could you show me this action, please ?

This point is fundamental, as the section 5 remains on buried structures at the origin of the GPR echoes.

I join the comments of the Reviewer 1, and would appreciate some complementary conclusions from IR+GPR

---

## Author Comment (AC2) · 15 Sep 2016

The reviewer is right in deeming that reflections in air are well matched with the anomaly in fig. 9, but this does not arease some "problematicity" of that anomaly. In fact, as the reviewer can see in the attahced file, the matching is achieved not with a hyperbola but rather with a couple of hyperbolas in air. The hypothesis of two pipes in the wall (they would have been expected quite shallow, so that the propagation of the waves would have been occurred substantially in air) was compatible (in principle) with these hyperbolas, even if the strenght of their crossing point (even hypothesizing some in-phase summation of the contributions) was anomalous. Moreover, indeed the best matching seems to us to be achieved for a propagation velocity of 27 cm/ns instead of 30, but this minimal difference might be due to some bent propagation path of the waves. In particular, we suspect that the spurious radiation of the antennas in

air is first of all "lateral", otherwise the target (that we now assume to correspond to the lights) should have appeared as a unique hyperbola with the maximum at the center of the Bscan.

Please also note the supplement to this comment:
http://www.geosci-instrum-method-data-syst-discuss.net/gi-2016-14/gi-2016-14-AC2-supplement.pdf

**Supplement:**

**v= 0.3 m/ns**

[Figure]

v= 0.29 m/ns

[Figure]

**v= 0.27 m/ns**

[Figure]

v= 0.25 m/ns

[Figure]

**v= 0.22 m/ns**

[Figure]

v= 0.21 m/ns